# Immune System Stimulation Reduces the Efficiency of Whole-Body Protein Deposition and Alters Muscle Fiber Characteristics in Growing Pigs

**DOI:** 10.3390/ani9060323

**Published:** 2019-06-06

**Authors:** Whitney D. McGilvray, Bradley Johnson, Hailey Wooten, Amanda R. Rakhshandeh, Anoosh Rakhshandeh

**Affiliations:** 1Department of Animal and Food Sciences, Texas Tech University, Lubbock, TX 79409, USA; whitney.mcgilvray@actx.edu (W.D.M.); bradley.johnson@ttu.edu (B.J.); hailey.wooten@ttu.edu (H.W.); 2Department of Biology, South Plains College, Levelland, TX 79336, USA; arakhshandeh@southplainscollege.edu

**Keywords:** immune system stimulation, muscle fibers, pigs, protein turnover

## Abstract

**Simple Summary:**

Disease reduces growth and protein retention in pigs. Protein retention is the balance between two energy-consuming processes in the body of pigs: protein synthesis and breakdown. Previous reports on the effects of disease on these components of protein metabolism and their consecutive effects on protein retention are inconsistent. In addition, limited information is available about the effects of disease on the composition of muscle fibers in pigs. Thus, we evaluated these parameters, since they help us to understand protein metabolism during disease in pigs. We used twelve gilts; five were used as a control and seven were made ill. Experimental diets were designed to supply nutrients that closely met the daily requirements of each group. Protein synthesis, protein breakdown, and protein retention were measured over 72 h. Pigs were then euthanized and various muscles were sampled. Our findings suggested that disease not only reduces protein retention by decreasing protein synthesis and protein breakdown, but also by reducing the efficiency of protein deposition. In other words, ill pigs synthesize more protein per unit of protein retention, compared to healthy pigs. In addition, disease reduces muscle mass and changes the composition of the muscle fibers. The latter might negatively affect pork quality.

**Abstract:**

The effects of immune system stimulation (ISS), induced by repeated injection of *Escherichia coli* lipopolysaccharide, on the whole-body protein synthesis versus degradation rates, the efficiency of protein deposition (PD), and muscle fiber characteristics in pigs were evaluated. Twelve growing gilts were assigned to two levels of amino acid intake that was predicted based on the potential of each group’s health status for PD and feed intake. Isotope tracer, nitrogen balance, and immunohistochemical staining techniques were used to determine protein turnover, PD, and muscle fiber characteristics, respectively. Protein synthesis, degradation, and PD were lower in immune-challenged pigs than in control pigs (*p* < 0.05). Strong tendencies for a higher protein synthesis-to-PD ratio (*p* = 0.055) and a lower protein synthesis-to-degradation ratio (*p* = 0.065) were observed in immune-challenged pigs. A decrease in muscle cross-sectional area of fibers and a shift from myosin heavy chain (MHC)-II towards MHC-I fibers (*p* < 0.05) were observed in immune-challenged pigs. These results indicated that ISS reduces PD not only by suppressing the whole-body protein synthesis and degradation rates, but also by decreasing the efficiency of PD in growing pigs. In addition, ISS induces atrophy in skeletal muscles and favors a slow-twitch oxidative fiber type composition.

## 1. Introduction

Immune system stimulation (ISS) and the subsequent cascade of events that occur during an immune response alter protein and amino acid (AA) metabolism, which often leads to reduced whole-body protein deposition (PD). Whole-body PD is positively correlated with growth performance and productive efficiency of growing pigs, and reflects the balance between whole-body protein synthesis and degradation [1,2,3,4,5,6]. Reports on the effects of ISS on the whole-body protein turnover rate and efficiency (i.e., synthesis + degradation) in pigs are inconsistent. Rudar et al. [7] reported a decrease in protein synthesis and the protein synthesis-to-PD ratio, an indicator of the efficiency of PD. They also reported no change in protein degradation in ISS pigs, compared to pair-fed healthy pigs. In contrast, Daiwen et al. [8] reported a reduction in protein degradation and the efficiency of PD, but no change in protein synthesis in immune challenged pigs, compared to their pair-fed, control counterparts [7,8]. These inconsistencies can largely be attributed to the nutritional strategies that were used in the studies, since the rate of protein turnover in the tissues and at the whole-animal level is highly influenced by diet composition and dry matter intake (DMI) [4,9,10,11]. Importantly, in both of these studies, animals in different ISS states were fed the same experimental diet under the assumption that both ISS groups had the same protein and energy requirements. However, this assumption is inaccurate and pair-feeding leads to an altered physiological state that suppresses whole-body protein turnover and alters the efficiency of dietary protein utilization in healthy, growing pigs [1,2,9,10,11]. Thus, information on the effects of ISS on whole-body protein turnover, especially the efficiency of it, in growing pigs that are fed to their potential for growth is lacking.

Previous studies with rats and pigs have suggested that ISS reduces protein synthesis and increases protein degradation in skeletal muscles [5,9,12]. Reduced protein synthesis and enhanced protein degradation together may result in net protein loss, and thus, a putative change in muscle fiber type composition during ISS, since these changes correlate with one another in pigs and other species [13]. Potential changes in muscle fiber type composition are important, since the morphological differences between fiber types, such as the cross-sectional area of fibers (CSAF) and the total number of fibers, are major determinants of muscle mass and pork quality. Metabolic and contractile features of muscle are also different between the muscle fiber types and can affect pork quality [14,15]. To our knowledge, no study has yet evaluated the effects of ISS on the morphological traits and the contractile and metabolic properties of muscle fibers in pigs. Therefore, the goal of this study was to measure the rate of whole-body protein turnover and efficiency of PD in immune challenged and healthy control pigs when they are fed according to their potential for PD and feed intake (FI). In addition, the effects of ISS on the fiber characteristics of various muscles were evaluated to better understand the relationship between muscle biology and protein turnover in different health states. 

## 2. Materials and Methods 

All methods and procedures for this experiment were reviewed and approved by the Texas Tech University (TTU) Animal Use and Care Committee (ACUC approval number 16038-05). All animal trials were conducted at the TTU Swine Research Center (New Deal, TX, USA).

### 2.1. General Design, Housing, and Treatments 

A total of twelve PIC gilts (Pig Improvement Company North America, Hendersonville, TN, USA) from three litters (body weight, BW, 31 ± 4.8 kg) were obtained from the TTU breeding herd and housed individually in stainless steel adjustable metabolism crates (1.7 × 0.7 m, with the tenderfoot flooring) in an environmentally controlled facility (temperature 18–22 °C). Pigs were allotted to treatment groups, such that at least one representative from each litter was found in each treatment group. Following a 4-day acclimatization to the experimental diets and environment [16], and 2 h prior to ISS, pigs were administered a single oral dose of ^15^N-glycine (10 mg/kg BW; glycine ^15^N, 98%, Cambridge Isotope Laboratories, Inc., Tewksbury, MA, USA). Immune system stimulation was induced by intramuscular (i.m.) injection of increasing amounts of *Escherichia coli* lipopolysaccharide (LPS; ISS+; 25 and 35 µg/kg BW; LPS strain 055:B5; Sigma Aldrich, St. Louis, MO, USA; *n* = 7). Pigs in the control group received sterile saline i.m. to account for the stress induced by injection (ISS−; *n* = 5). Injections were given 48 h apart and whole-body nitrogen (N) flux was measured for 72 h using the end-product method, following the first injection of LPS [17]. The initial dose of LPS was increased by 29% for the subsequent injection to overcome tolerance to LPS. It has been previously demonstrated that this model of ISS induces a mild immune response with moderate clinical symptoms, such as a low degree fever and mild to moderate lethargy [18].

Pigs were assigned to two levels of standardized ileal digestible (SID) amino acid (AA) intake, which was predicted based on the potential of each health status group for protein deposition (PD) and DMI using NRC Swine software [3]. Blood samples were collected 24 h post ISS by jugular venipuncture and assayed for measures of blood chemistry. Infrared (IR) thermography was performed to monitor body temperature (BT) during the course of the study, as described by Petry et al. [19]. At the end of the study, pigs were euthanized by intravenous (i.v.) injection of a lethal dose of sodium pentobarbital (FATAL PLUS, Vortech Pharmaceutical, Ltd., Dearborn, MI, USA). Immediately after euthanasia, muscle samples were taken and processed for determination of fiber type composition and CSAF.

### 2.2. Experimental Diets and Feeding

For each ISS group, diets were formulated based on the nutrient requirements that were predicted using the NRC Swine model and based on performance variables determined in previous studies [1,2,3,20]. The performance variables included a mean initial BW of 25 kg, an average daily feed intake (ADFI) of 1.3 kg/day for ISS− and 0.9 kg/day for ISS+, and PD of 100 and 60 g/day for ISS− and ISS+, respectively [3]. Within each ISS group, levels of dietary SID essential AA were adjusted to meet the potential of each ISS group for optimum PD. A constant ratio among SID crude protein (Cp), SID AA and SID lysine was maintained across both experimental diets (Table 1). All experimental diets were isoenergetic and contained 14 MJ/kg of metabolizable energy. The diet of ISS+ pigs was formulated to have a higher metabolizable energy (ME)-to-lysine ratio to minimize the impact of energy intake on components of whole-body protein turnover and PD [21]. All experimental diets were formulated to supply a minimum of 2.2 times the energy requirements for maintenance. The experimental diets were fortified with vitamins and minerals to surpass the requirements recommended by NRC Swine [3]. Titanium dioxide (0.25% TiO_2;_ Bobette Boyer Hall Technologies, St. Louis, MO, USA) was included in all diets as an indigestible marker to determine nutrient digestibility (Table 1). Pigs were fed either 1250 or 850 g/day divided into two equal meals. The pigs were fed twice per day, and were allowed free access to water. 

### 2.3. Observations and Sampling 

Body temperature (BT), blood chemistry, feed waste, and vomit were measured, as previously described [19,22]. Briefly, eye temperature was measured at times 0, 2, 4, 6, 8, 10, 24, 48, 50 and 72 h post LPS injection using a FLIR E40 digital camera (FLIR Systems, Inc., Wilsonville, OR, USA), and measures of blood chemistry, hematology and acid/base balance were determined using an i-STAT Handheld Analyzer (Abaxis Inc., Union City, CA, USA) with an i-STAT CHEM8+ cartridge. Feed waste and vomit for each pig was collected in trays beneath each feeder, dried, and weighed to accurately determine daily feed and N intake. Urine was collected according to the procedures described previously [22]. In brief, total urinary output was collected in six consecutive 12-h periods, for a total of 72 h, to determine urinary ammonia and urea N and ^15^N excretion. We chose this timeframe since previous studies found that more than 95% of the total urinary ^15^N excretion is in the form of urea and ammonia within 60 to 72 h of oral administration of ^15^N-glycine [17,23]. Urine was collected via collection trays underneath each crate that funneled urine into tared, lidded buckets containing sufficient amounts of 3 N HCl to maintain urine pH below three. Urine samples were collected 12 h prior to the oral administration of ^15^N-glycine to determine natural abundance of ^15^N in urinary ammonia and urea. Briefly, for each 12-h collection, urine was weighed and a 10% aliquot was sampled, placed in a sealed collection cup, and stored at 4 °C before further analysis. At the end of the study, samples were pooled for each pig to represent the combined sampling periods from 12 to 72 h after oral ^15^N-glycine administration, so that ^15^N enrichment in the urine could be determined.

The impacts of ISS on muscle fiber characteristics were determined by collecting muscle samples (MS) from the *longissimus dorsi* (LD; right LD taken at the 8th to 9th thoracic vertebrae), right *serratus ventralis* (SV), *semitendinosus* (ST; right ST taken approximately 1cm from the tendon attachment point on each end), and right *psoas major* (PM) immediately following euthanasia. Following sample collection, muscle fiber orientation was identified and muscles were sectioned in the center of the muscle into a 2 × 2 × 3 cm portion, placed into a mold, and embedded in Clear Frozen Section Compound (VWR International, Randor, PA, USA). The samples were frozen using dry ice-chilled 2-methyl-butane and placed in a cooler of dry ice for transport [24]. Samples were then stored at −80 °C until further analysis. 

### 2.4. Analytical Procedures

All analyses were performed in duplicate unless otherwise stated. Urinary urea N concentration was determined for each pooled urine sample using a colorimetric method and a commercial kit, according to the manufacturer’s procedures (QuantiChrom Urea Assay Kit, Bioassay Systems, Hayward, CA, USA). Urinary ammonia N concentration was measured using a commercial kit and according to the manufacturer’s instructions (EnzyChrom ammonia/ammonium Assay Kit, Bioassay Systems, Hayward, CA, USA). An adaptation of the method described by Rivera-Ferre et al. [17], was followed with some modifications. Briefly, for the isolation of ammonia and urea, the pH of the centrifuged urine samples was adjusted to between five and six using 10 M NaOH [25]. An aliquot of the urine containing approximately 3000 µg ammonia N was then passed through a chromatography column (Poly-Prep 731-1550, BioRad, Hercules, CA, USA) containing 0.5 mL of a cation exchange resin (Dowex 50W X8 H^+^ form, 200–400 mesh, Sigma-Aldrich, St. Louis, MO, USA) that was converted to the Na/K form. The eluent contained the urea fraction. Ammonia bound to the column was then eluted by adding 1 mL of 1M KOH to the column and collected into a solution containing 35 µL of 12 M sulfuric acid and 20 µL bromphenol blue (1% in Milli-Q water). Samples were then stored at −20 °C until further analysis. The elute containing the urea fraction was then subjected to enzymatic hydrolysis using a urease solution containing 400 units/mL suspended in a sodium phosphate buffer, pH 7 (Urease Type C-3 Jack Bean, ≥600,000 units/g solid, Sigma-Aldrich, St. Louis, MO, USA). A portion of the eluent, containing approximately 3000 µg of urea N, was added to a 10 mL Vacutainer (BD Vacutainer, Franklin Lakes, NJ, USA). Vacutainers were recapped and 100 µL of a urease solution was injected through the rubber stopper. Samples were then incubated at 30 °C for 1 h in an inverted position to ensure there was no ammonia loss. Following incubation, 0.5 mL of 1 M HCL was injected through the stopper to end the reaction. The solution was then passed through the chromatography columns, as previously described for ammonia N extraction, to obtain the urea N fraction [17]. Lastly, both ammonia N and urea N fractions were prepared for combustion in an elemental analyzer. Briefly, 35 µL of each sample was absorbed in 2 mg Chromosorb W (30–60 mesh Acid Washed 10 gm; Elemental Microanalysis, Okehampton, UK) in tin capsules (Elemental Microanalysis, Okehampton, UK). Isotopic enrichment of urinary ammonia and urea N was measured by the University of California Davis Stable Isotope Facility using an Elementar Vario Micro Cube elemental analyzer (Elementar Analysensysteme GmbH, Hanau, Germany) interfaced to a PDZ Europa 20-20 isotope ratio mass spectrometer (Sercon Ltd., Cheshire, UK). The ^15^N enrichment in each fraction was obtained after correcting for the natural abundance of ^15^N in urine.

Preparation of embedded muscle samples for immunohistochemical sectioning and staining were carried out as described by Hergenreder et al. [24]. In brief, embedded muscle samples were transferred from −80 °C to −20 °C for 24 h prior to sectioning. Embedded samples were then cut into 10-µm cross sections at −20 °C using a Leica CM1950 cryostat (Lieca Biosystems, Buffalo Grove, IL, USA). Two cryosections were mounted on positively charged glass slides (Superfrost Plus; VWR International, Radnor, PA, USA) for the determination of muscle fiber distribution, area, and nuclei density. Immunohistochemical staining of cross sections was carried out as described by Hergenreder et al. [24] with the exception of the antibodies used in the current study. Cryosections were incubated in the following primary antibodies: a 1:100 volume to volume (vol:vol) ratio of anti-myosin heavy chain (MHC)-IIB IgM (10F5 Supernatant), a 1:75 (vol:vol) ratio of anti-MHC-IIA and anti-MHC-I IgG1 (BF-35 Supernatant), and a 1:100 (vol:vol) ratio of anti-MHC-I IgG2b (BA-D5 Supernatant). Antibodies were purchased from Developmental Studies Hybridoma Bank (University of Iowa, Iowa City, IA, USA). After rinsing each slide three times in a phosphate buffer solution (PBS), cryosections were incubated for 30 min at room temperature in opaque boxes in the following secondary antibodies: a 1:1000 (vol:vol) ratio of goat α-mouse IgM, Alexa-Fluor 488; a 1:1000 (vol:vol) ratio of goat α-mouse IgG1, Alexa-Fluor 546; and a 1:1000 (vol:vol) ratio of goat α-mouse IgG2b, Alexa-Fluor 633. All secondary antibodies were purchased from Invitrogen (Carlsbad, CA, USA). After rinsing each slide three times in PBS, cover-slips were added with ProLong Gold mounting media (Thermo Fisher Scientific, Waltham, MA, USA) containing 4’6-diamidino-2-phenylindole (DAPI) and allowed to cure horizontally in opaque boxes for 36 h at room temperature. Images were taken within 48 h of curing, as described by Hergenreder et al. [24]. All slides were imaged at a 200X working difference magnification using an inverted fluorescence microscope (Nikon Eclipse, Ti-E; Nikon Instruments, Inc., Mellville, NY, USA) with a UV light source (Intensilight C-HGFIE; Nikon Instruments, Inc., Mellville, NY, USA) and a CoolSnap ES^2^ monochrome camera (Photometrics, Tucson, AZ, USA). Images were artificially colored and analyzed using NIS Elements Imaging software (Nikon Instruments, Inc., Mellville, NY, USA). Five random images were taken of each slide fixed with two cryosections (Appendix A). For each image, all MHC-I, MHC-IIA, MHC-IIB, and MHC-IIX fiber types were identified and expressed as a percentage of the total fiber number. Type IIX fibers were identified on the basis of a lack of staining with antibodies directed against type I, IIA, and IIB MHC. The cross-sectional area of each fiber type in each image was measured using the NIS Elements software (Nikon Instruments Inc., Mellville, NY, USA). Nuclei density was determined by counting the total number of DAPI-stained cells in each image. 

### 2.5. Calculations and Statistical Analysis

A power test was used to determine the number of animals per ISS group. The coefficient of variance (%) was five, the percent difference from the control was decided at 10, and *p* ≤ 0.05 with a power of 90%. Whole-body protein turnover was determined using the end-product method [26], using a single oral dose of ^15^N-glycine. This method assumes that the ^15^N from isotopically labeled glycine is partitioned between protein synthesis and AA oxidation in the same proportion as unlabeled AA, and that the ^15^N released from protein degradation is not re-incorporated into whole-body protein during the collection period. Based on these assumptions, the proportion of ^15^N excreted in the end-product (urinary ammonia or urea) compared to the total dose given reflects the contribution of unlabeled N excreted as that end-product compared to the total flux. Thus, Equation (1) was used [17,26]:*d*/*ex* = *Q*/*Ex*(1)
where *d* is the dose of ^15^N given orally (g), *ex* is the total amount of ^15^N excreted as urinary ammonia or urea (g) during the collection period, *Q* is the total whole-body N flux (g/d), and *Ex* is the amount of N excreted in urinary ammonia or urea (g/d).

Rates of whole-body N flux were calculated separately for urinary ammonia (Q_A_) and urea (Q_U_) as end products, since they often have different ^15^N enrichment. Rates were then estimated by the arithmetic mean [26], using Equation (2) [27]:*Q* = *Ex* (*d*/*ex*)(2)

Absolute rates of whole-body protein synthesis and degradation were estimated from Equation (3):*Q* = *I* + *B* = *S* + *U*(3)
where *Q* represents the total whole-body nitrogen flux (g/kg BW^0.60^/day), *I* is the rate of dietary nitrogen entering the pool (g/kg BW^0.60^/day), *B* is the rate of whole-body protein breakdown (g N/kg BW^0.60^/day), *S* is the rate of whole-body protein synthesis (g N/kg BW^0.60^/day), and *U* is the total urinary nitrogen excretion (g/kg BW^0.60^/day).

Statistical analyses were carried out using SAS software version 9.4 (SAS Institute, Cary, NC, USA). Normality and homogeneity of variances were confirmed using the univariate procedure (PROC UNIVARIATE). Outliers were determined as any value that differed from the treatment mean by ± 2 standard deviations. Data were analyzed in a complete randomized design using mixed procedures (PROC MIXED). Health status (ISS) and pig within ISS were used as fixed and random effects, respectively, when data on whole-body N metabolism were analyzed. Nuclei density and muscle fiber type data were analyzed with ISS and the specific muscle sampled (MS) and the interaction between ISS and MS (ISS × MS) as fixed effects, and pig within ISS (treatment) as a random effect. Fiber area data were analyzed with ISS, MS, and fiber type and the interaction among the main effects as fixed effects, and pig within ISS (treatment) as a random effect. For parameters such as BT that were measured over time, repeated measurements analysis of variance was used. An appropriate covariance structure was selected for analyses by fitting the model with the structure, which provided the ‘best’ fit, based on Akaike information criterion and Schwarz Bayesian criterion. Means were separated using the Tukey–Kramer test. Treatment effects were considered significant at *p* ≤ 0.05. A tendency towards a significant difference between treatment means was considered at *p* ≤ 0.10. 

## 3. Results

### 3.1. General Observations

All pigs readily consumed experimental diets and showed signs of good health prior to the study. Pigs challenged with LPS displayed clinical symptoms of disease, such as lethargy, fever, and vomiting. However, ISS did not affect the intake of the daily feed allowance during the post-ISS period. Data from one pig was excluded from the study due to a severe immune response to the LPS injection. Analyzed dietary nutrient contents were generally in agreement with calculated values (± 5%) that were derived from feed ingredient composition and nutrient levels in feed ingredients according to the NRC Swine (Table 1) [3].

### 3.2. Body Temperature, Hematology and Blood Chemistry

Immune system stimulation increased eye temperature by 0.6 ± 0.12 °C (38.4 vs. 39.0 °C, SE 0.08; *p* < 0.001). Relative to ISS− pigs, ISS+ pigs had higher blood urea nitrogen (BUN; 4.6 vs. 24.1 mg/dL, SE 7.60; *p* < 0.05) and creatinine (1.0 vs. 3.0 mg/dL, SE 0.70; *p* < 0.050) levels, as well as anion gap (AnionGAP; 20.5 vs. 22.8 mEq/L, SE 0.60; *p* < 0.041). Hemoglobin (Hb; 10.8 vs. 9.5 g/dL, SE 1.19), hematocrit as the percent of packed cell volume (HCT; 31.8 vs. 27.9%, SE 3.51), and blood glucose (96.8 vs. 100.7 mg/dL, SE 5.0) levels were not affected by ISS (*p* > 0.330). 

### 3.3. Whole-body N Metabolism

The measures of whole-body N metabolism in ISS- and ISS+ groups are presented in Table 2. Final BW was lower in the ISS+ group compared to the ISS− pigs (*p* < 0.018). As anticipated, ISS+ pigs had a lower dietary N intake relative to ISS− pigs (*p* < 0.001). However, no differences between ISS groups were observed for urinary N excretion (*p* = 0.45), N excretion via urinary ammonia (*p* = 0.81), or N excretion via urinary urea (*p* = 0.413). Excretion of ^15^N via urea was not affected by ISS at the end of the 72-h collection period (*p* = 0.110). However, a tendency for higher ^15^N excretion via ammonia was observed in ISS+ pigs compared to ISS− pigs (*p* = 0.101; Figure 1). As anticipated, whole-body N flux, protein synthesis and protein degradation (g N/kg BW^0.60^/day), as well as protein retention and N-balance, were lower in the ISS+ group compared to ISS− pigs (*p* < 0.001). The nitrogen retention-to-N intake ratio (N-Retention: N-Intake) was 51% lower in the ISS+ group compared to ISS− pigs (*p* < 0.033). Also, a strong tendency was observed for a lower protein synthesis-to-degradation ratio in ISS+ pigs compared to ISS− pigs (S:D; *p* = 0.065). In addition, the protein synthesis-to-retention ratio, a measure of the efficiency of PD, tended to be higher in ISS+ pigs than in ISS- pigs (S:R; *p =* 0.055). 

### 3.4. Muscle Fiber Characteristics

Data on nuclei density and muscle fiber type distribution in the LD, PM, ST, and SV are presented in Table 3. An interaction effect between health status and the muscle sampled (ISS × MS) was observed for nuclei density (*p* < 0.001). In the PM, ISS significantly reduced the density of nuclei (*p* < 0.001), but had no effect on this parameter in other muscle types. An interaction effect between ISS and MS was also observed for the percentage of MHC-I and MHC-IIX fiber types (*p* < 0.050), but not for MHC-IIA fibers (*p* = 0.164). Relative to ISS− pigs, ISS+ pigs had a higher percentage of MHC-I in the LD and SV (*p* < 0.01), while the MHC-I percentage was not affected in the PM (*p* > 0.200). Similarly, ISS reduced the percentage of MHC-IIX fibers in the LD and SV muscles (*p* < 0.020), but had no effect on this muscle fiber type in the PM and ST (*p* > 0.530). The percentage of MHC-IIA fibers was reduced in the ST (*p* < 0.050) and tended to be reduced in the PM (*p* = 0.089) of ISS+ pigs relative to ISS− pigs, while it was not affected in the LD and SV (*p* > 0.800). The MHC-IIB fiber type was only detected in the SV and LD muscles of two pigs in the ISS− group. Figure 2 demonstrates the effects of treatments on CSAF. The effects of ISS on CSAF varied among the different muscles sampled, leading to a significant interaction between ISS, MS, and fiber type for fiber cross-sectional area (*p* < 0.001). Immune system stimulation reduced CSAF of MHC-I fibers in the LD and ST (*p* < 0.050), but increased the MHC-I CSAF in the SV (*p* < 0.050). Immune system stimulation reduced MHC-IIX CSAF in the LD, ST, and SV (*p* < 0.050), but increased it in the PM (*p* < 0.001). Lastly, ISS significantly reduced the MHC-IIA CSAF in the ST (*p* < 0.001), but had no effect in the LD, PM and SV. 

## 4. Discussion

The main objective of the current study was to evaluate the immunopathology caused by the immune response on the efficiency of whole-body protein turnover and skeletal muscle fiber characteristics in growing pigs when they are fed according to their potential for growth. Immune system stimulation is known to reduce PD in pigs, and thus, the dietary requirements for AA [1,2,18,28,29]. Thus, in the current study, the AA contents of the diets for each ISS group were adjusted based on the potential of each ISS group for PD, since PD is the main determinant of daily protein and energy requirements in growing pigs, even during ISS [1,2,3,5]. We chose this nutritional strategy to avoid the confounding effects of feeding excessive AA on whole body protein turnover in ISS pigs [30]. Furthermore, in the present study, we did not pair-feed the control (ISS-) animals, since restricted feed intake can downregulate the rate of whole-body protein turnover and alter the efficiency of dietary N and AA utilization in non-immune challenged pigs [3,26,30]. Instead, in the present study, the daily feed allowance for the ISS+ pigs was determined using previous studies that observed a constant quantitative decrease in feed intake with the same model of ISS [1,2,18,29,31]. 

In the current study, ISS was induced by repeated injections of increasing amounts of LPS. We previously have shown that this model of ISS induces a relatively moderate, but effective, ISS that can simulate the immunopathology caused by moderate clinical diseases [1,2,18]. In the present study, this ISS model resulted in a febrile response, as determined by eye temperature. Eye temperature remained elevated for the entire duration of the study in ISS pigs, suggesting a lasting elevated core BT, an indicator of a systemic inflammation [19]. The pro-inflammatory cytokine interleukin-1β serves as the main endogenous pyrectic substance in initiating fever and works synergistically with interleukin-6 and TNF-α [6]. Therefore, an increase in BT suggests that ISS in the current study has been modulated by pro-inflammatory cytokines. Furthermore, in the present study we observed an increase in AnionGAP in ISS+ pigs compared to the ISS− pigs. This finding likely occurred because of an increased level of lactic acid in the blood, since we previously showed that the electrolyte balance, determined by Na^+^, K^+^, and Cl^−^ concentrations, is not affected by the model of ISS used here [31]. Therefore, increased AnionGap in ISS+ pigs may reflect a shift from aerobic to anaerobic glycolytic metabolism that usually occurs during the acute phase response of systemic inflammation [32], providing further support for effective ISS. Long term and severe inflammatory responses in various species often result in reduced Hb levels and HCT. The latter occurs because of pro-inflammatory cytokine mediated interference with reticuloendothelial iron transport, decreased sensitivity of erythron to erythropoietin, and reduced erythrocyte survival during severe inflammation [33]. The lack of an effect of ISS on Hb levels and HCT in our study is likely because our model of ISS stimulated only a mild inflammatory response. Metabolic changes during ISS are often characterized by reduced blood glucose levels during the post-absorptive state, due to enhanced glucose uptake by immune cells as their preferred source of energy [34]. However, in the current study, ISS had no effect on blood glucose levels, since our pigs were in the fed state when blood samples were taken. Taken together, these results indicated that repeated injection of increasing amounts of LPS induced an effective ISS in our study. 

Measures of whole-body protein retention, which are determined by conventional N-balance methods, do not allow for individual estimates of the contribution of protein synthesis, protein degradation, and the efficiency of these processes for whole-body protein gain. Therefore, in the present study, we used the end-product method and a single bolus dose of ^15^N-glycine to directly measure these aspects of whole-body protein turnover in growing pigs. This simple and non-invasive method provides more precise estimates of whole-body N flux and protein turnover, than the estimates obtained when uniformly labeled protein is used [26]. However, a limitation of this technique is that it cannot determine the contribution of individual tissues or protein pools to whole-body protein turnover [4]. In this study, N-retention, calculated as the balance between protein synthesis and protein degradation, was in close agreement with the N-balance measurements taken simultaneously in the same pigs, providing confidence that our measures of protein synthesis and degradation were precise, despite potential technical limitations. Similarly, the observed estimates of whole-body N flux in the current study were comparable to the estimates in pigs by other workers [7,8,11,17].

Reduced protein gain and increased N loss, due to reduced feed intake and efficiency of AA utilization for PD, are characteristics of an immune response [5]. Distinct changes in whole-body protein and AA metabolism during ISS are multifaceted and include changes in both protein synthesis and degradation in disparate tissues. The contributions of the main protein pools (i.e., skeletal muscle and visceral tissues) to whole-body protein turnover can be highly influenced by age, as well as the nutritional or physiological state of an animal. For instance, the contribution of muscle tissue to whole-body protein turnover is much greater than that of the visceral tissues in growing animals. However, the contribution of muscle tissue declines as the animal ages [12,13,35,36,37]. Although in the current study the diets were designed to supply sufficient AA for optimum PD in each ISS group, the efficiency of dietary N utilization for PD, as determined by the N retention-to-N intake ratio, was significantly lower in ISS+ pigs. This was likely due to a higher rate of AA catabolism in immune challenged pigs. These results were consistent with the enhanced BUN (the primary metabolite derivative of AA catabolism) and blood creatinine levels (an indicator for increased skeletal muscle degradation) that were seen in the present study [38]. In addition, in this study the total urinary N excretion did not differ between ISS groups, even though the N intake was lower in ISS+ pigs than in ISS− pigs. This result also suggested a higher rate of AA catabolism in ISS+ pigs, since the N derived from AA catabolism is mainly excreted
via urine [20]. Furthermore, the observed tendency for higher ^15^N excretion via urinary ammonia in ISS+ pigs can likely be associated with a higher rate of AA catabolism in the peripheral tissues (i.e., skeletal muscle) of ISS+ pigs. This idea is supported by observations that urinary ammonia is mainly derived from glutamine, which is synthesized predominantly in the muscle to transport excess N [39]. Importantly, higher ^15^N excretion via ammonia in ISS+ pigs did not affect the estimate of whole-body N flux, since the urinary ammonia-to-urea ratio was not different between ISS groups [26]. Moreover, we observed a strong tendency for a reduced whole-body protein synthesis-to-degradation ratio in ISS+ pigs, which further indicated a catabolic state in immune-challenged pigs compared to their unchallenged counterparts [4]. Together, these results indicated that ISS induced by repeated injection of increasing amounts of LPS negatively affected the efficiency of N utilization in growing pigs. The findings of the current study are in general agreement with those of Ruder et al. [7] and Daiwen et al. [8], who reported similar findings in LPS-challenged pigs.

Protein turnover is one of the most energy consuming processes in the body of animals, and it determines the efficiency of protein and AA utilization [10,11]. In the current study, the rate of whole-body protein synthesis was lower in ISS+ pigs, which can most likely be associated with the reduced protein synthesis seen in skeletal muscle. Reportedly, ISS increases both the fractional and absolute synthesis rates of protein in the visceral tissues, especially in the liver, while reducing protein synthesis in the skeletal muscles. This shift is thought to predominantly occur due to (i) reduced FI, and (ii) cytokine-mediated reduced protein synthesis in skeletal muscle [13,35,37,40]. The results of the current study, however, suggest that the reduction in protein synthesis in the skeletal muscles of growing pigs during ISS was much greater than the increase in protein synthesis in the visceral tissues. Considering that skeletal muscle is the major protein pool contributing to whole-body protein turnover in growing pigs, increased protein synthesis in the visceral tissues was probably masked by reduced protein synthesis in the skeletal muscles when whole-body protein synthesis was determined in ISS+ pigs [36,37]. The findings of the present study agree with those of Daiwen et al. [8], who reported a substantial decrease in whole-body protein synthesis of ISS pigs, compared to the healthy control pigs whose feed intake was not restricted. In the latter study, however, whole-body protein synthesis did not differ between ISS pigs and pair-fed healthy pigs, most likely because protein synthesis was limited by energy intake in the pair-fed healthy pigs [3,21]. The findings of the current study are also in general agreement with the findings of Rudar et al. [7], who reported a decrease in whole-body protein synthesis of feed-restricted pigs 36 h after LPS challenge. In the present study, the rate of whole-body protein degradation was lower in ISS pigs, most likely due to lower dietary N intake, since the rate of whole-body protein degradation can be substantially downregulated by reduced feed intake alone [10,11]. In the Daiwen et al. study [8], LPS challenge resulted in lower protein degradation relative to non-feed restricted control animals, but higher protein degradation relative to pair-fed newly weaned pigs. In contrast, the Rudar et al. study [7] observed that LPS challenge had no effect on protein degradation in growing pigs. These discrepancies can most likely be associated with the nutritional strategies that were adopted in these studies. Nonetheless, in both of these studies ISS resulted in a lower efficiency of protein accretion, as determined by the protein synthesis-to-protein retention ratio, which is in agreement with our observations in the current study. Thus, from these findings it is evident that ISS not only suppresses the whole-body protein turnover rate, but it also reduces the efficiency of PD by increasing the rate of protein degradation relative to protein synthesis. 

In the present study, we used immunohistochemistry to evaluate the effects of ISS on muscle fiber characteristics, because it can distinguish between pure and hybrid fibers in swine. Thus, this technique is not only less laborious and more cost effective than other techniques, such as magnetic resonance imaging, phosphorus magnetic resonance spectroscopy, or the biochemical identification of metabolic enzymes, but it is also more accurate [41]. In the current study, we could identify MHC-IIB fiber types in the muscles of only two pigs, most likely because the transition of type II MHC fibers from type IIA and IIX to type IIB predominantly takes place as an animal approaches maturity [42]. Alternatively, the latter could be caused by a low positive reaction of the antibody that was used in the current study with MHC-IIB type fibers [43]. In the present study, the percentage of fast-twitching glycolytic fibers, predominantly MHC-IIX, was lower and the percentage of slow-twitching oxidative fibers (i.e., MHC-I) was higher in the ISS+ group compared to the ISS− group. These results were consistent with other findings in starter pigs, which reported that LPS-induced ISS reduced skeletal muscle protein synthesis, preferentially in fast-twitch glycolytic muscles relative to oxidative muscle types, compared to pair-fed healthy control pigs [13,15,40]. This change in muscle fiber type agrees with the observed decrease in CSAF, especially in large muscles, such as the LD and ST. Reduced CSAF in ISS pigs may also influence PD, since muscle fiber size affects muscle growth potential in pigs [44]. Although our study did not directly measure the rate of protein synthesis in skeletal muscle, the general decrease in nuclei density, combined with reduced CSAF in ISS pigs, suggests that skeletal muscle protein synthesis was reduced and may account for the reduced whole-body protein synthesis that was observed. Reduced protein synthesis in the muscle, and thus reduced muscle mass during ISS, may be explained, by i) ISS-induced inhibition of anabolic hormones, and ii) reduced FI. Indeed, fast-twitch glycolytic fiber types are particularly sensitive to insulin stimulated protein synthesis [45]. During ISS, pro-inflammatory cytokines, in particular TNF-α, inhibit the insulin receptor, which suppresses mTOR-dependent translation initiation [14] and inhibits glucose uptake. Thus, enhanced glucose utilization by immune cells can occur [34,46]. During ISS, pro-inflammatory cytokines also cause adipocyte lipolysis and increase the plasma levels of free fatty acids [47], which would support a predominantly slow-twitch oxidative fiber type composition. On the other hand, reduced FI during ISS might have played an additional role in the observed changes in muscle fiber characteristics. In the current study, since a pair-fed group was not included in our experimental design, we cannot make a clear distinction between the effects of immunopathology caused by ISS and that of reduced FI. Nonetheless, whether the observed changes in muscle fiber characteristics of ISS pigs were caused by the immunopathology of ISS and/or reduced FI, they can be associated with ISS, since reduced FI is an unavoidable component of an immune response [6]. Finally, the altered fiber type composition and reduced CSAF observed in ISS pigs in the present study could potentially have a negative impact on pork quality. For instance, an increased proportion of MHC-I fibers decreases meat color stability and causes a shift to a brownish color [48]. Furthermore, a strong positive correlation between CSAF and intramuscular fat content in porcine LD has been reported [49]. Thus, a reduction in CSAF, especially in MHC-I type fibers, might reduce the palatability of pork loin. Moreover, reduced CSAF may decrease pork tenderness, since a positive relationship between CSAF and tenderness has been reported in the muscles of other species [50]. Further investigation into the effects of ISS-induced changes in fiber type composition and CSAF on pork quality is warranted. Collectively, these results suggest an ISS-induced shift in fiber type composition toward a slow-twitch oxidative fiber type (MHC-I). Immune system stimulation also resulted in a decreased CSAF in both MHC-I and MHC-II fibers. These results can be attributed to an ISS-induced atrophy of skeletal muscle, which correlates with the ISS-induced decreases in whole-body protein synthesis, degradation, and retention that were observed here. 

## 5. Conclusions

Repeated injection with increasing amounts of *Escherichia coli* lipopolysaccharide (LPS) elicited an effective immune response that allowed the impact of immune system stimulation (ISS) on whole-body protein turnover and skeletal muscle biology to be evaluated. Collectively, our results indicate that ISS reduces protein gain in starter pigs, not only by reducing the whole-body protein synthesis and degradation rates, but also by reducing the efficiency of protein retention. Immune system stimulation reduces the whole-body protein turnover rate, mainly by reducing the voluntary feed intake, but also by reducing the protein synthesis rate more than the protein degradation rate. Immune system stimulation also caused muscle atrophy and a shift in fiber type composition, increasing the percentage of slow twitching fibers. Thus, our results suggest that reduced protein synthesis and retention at the whole-body level may occur primarily in the muscles during ISS. The energetic and nutrient costs associated with a reduced whole-body protein turnover rate and efficiency of protein deposition (PD) need to be considered when nutritional strategies are adopted for growing pigs during ISS. Thus, these results warrant further studies to evaluate the effects of ISS on tissue protein turnover, dietary energy requirements, and pork quality in pigs. 

## Figures and Tables

**Figure 1 animals-09-00323-f001:**
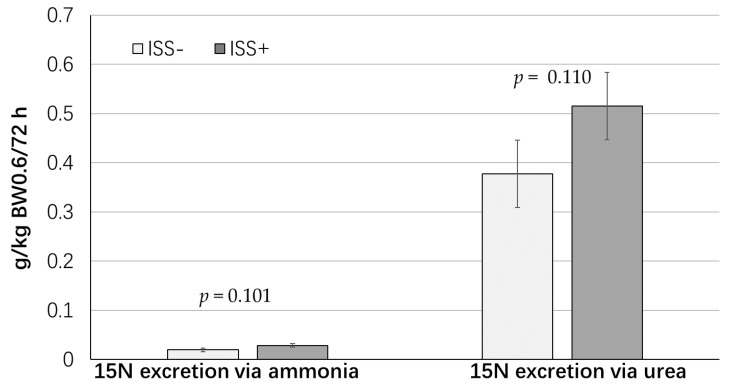
Effects of immune system stimulation (ISS) on cumulative urinary urea and ammonia ^15^N excretion in pigs 72 h after oral administration of ^15^N-glycine. Data are the least square means ± the largest standard error of mean (SE). Twelve gilts (initial BW 31 ± 4.8 kg) were subjected to one of two health states: immune system stimulated (ISS+; *n* = 7), or healthy control (ISS−; *n* = 5). Repeated i.m. injection of *E. coli* lipopolysaccharide (LPS; 25 and 35 µg/kg BW, given 48 h apart) was used to induce ISS. Pigs in the ISS− group received sterile saline i.m. Diets were formulated to meet daily standardized ileal digestible (SID) amino acid requirements for each ISS group, which were estimated based on the potential of each group for protein deposition according to the NRC Swine model [3].

**Figure 2 animals-09-00323-f002:**
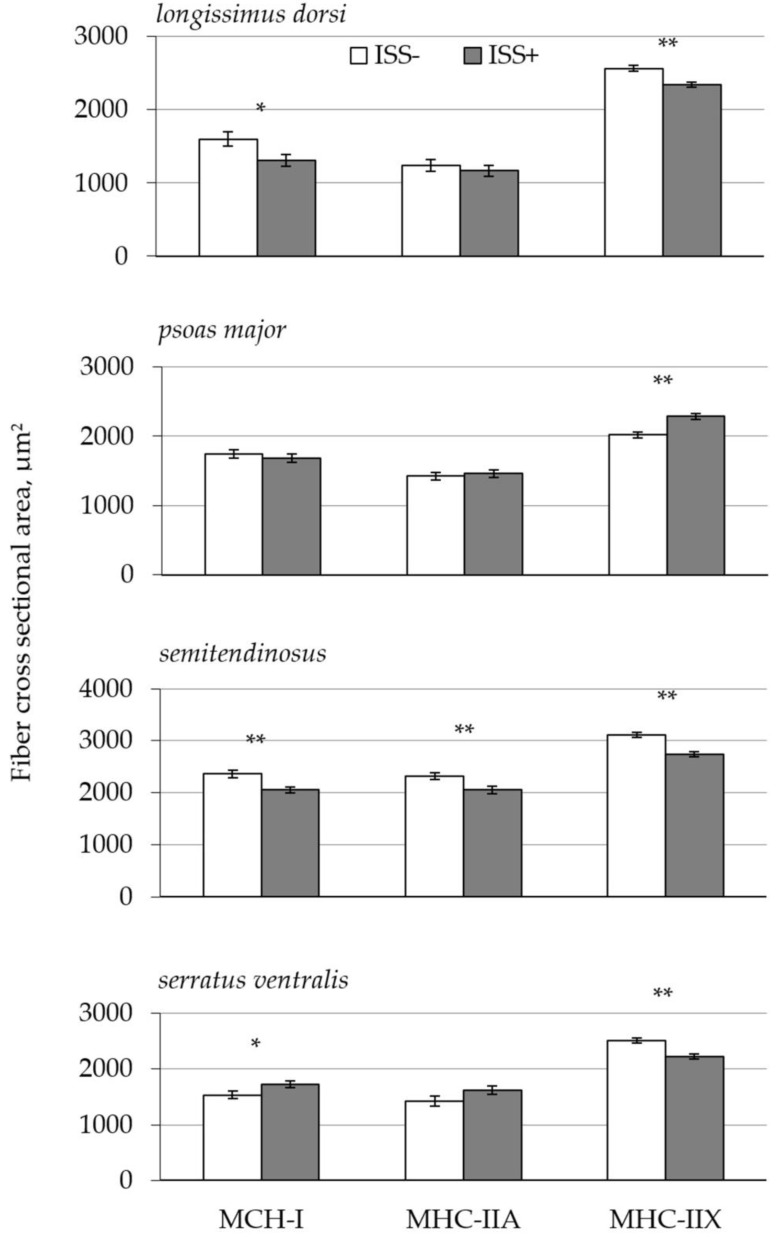
Cross-sectional area of muscle fibers (CSAF) in growing pigs with different health states. Data are the least square means ± the largest standard error of mean (SE). Twelve gilts (initial BW 31 ± 4.8 kg) were subjected to one of two health states: immune system stimulated (ISS+; *n* = 7), or healthy control (ISS−; *n* = 5). Repeated i.m. injection of *E. coli* lipopolysaccharide (LPS; 25 and 35 µg/kg BW, given 48 h apart) was used to induce ISS. Pigs in the ISS− group received sterile saline i.m. Diets were formulated to meet daily standardized ileal digestible (SID) amino acid requirements for each ISS group, which were estimated based on the potential of each group for protein deposition according to the NRC Swine model [3]. Pigs were euthanized 72 h post-ISS and muscle samples were collected immediately following euthanasia. *, *p* < 0.05; **, *p* < 0.01.

**Table 1 animals-09-00323-t001:** Ingredient composition and nutrient contents of experimental diets ^†.^

Ingredients and Nutrients Composition	Treatment	
	ISS−	ISS+
Ingredient composition, (g/kg as-fed basis)		
Corn	740	788
Soybean meal	206	159
Hydrogenated vegetable fat	10	10
Lysine HCl	4.1	3.8
DL-Methionine	1.0	0.73
L-Threonine	0.34	0.38
L-Tryptophan	0.2	0.25
Limestone	8.8	8.3
Dicalcium phosphate	7.9	7.4
Salt	5	5
Vitamin and mineral premix	15	15
Titanium dioxide	2.5	2.5
Calculated nutrient contents (g/kg as-fed basis)		
Metabolizable energy MJ/kg	14.0	14.0
Cp (N × 6.25) ^§^	140	121
Lysine	10.1	8.8
Methionine	3.3	2.9
Methionine + Cysteine	5.7	5.0
Threonine	5.1	4.5
Tryptophan	1.7	1.5
Leucine	12.9	11.8
Isoleucine	5.8	5.0
Valine	6.6	5.8
Phenylalanine	6.9	6.1
Calcium	6.1	5.6
STTD P *	2.9	2.7
Analyzed Cp and AA contents (g/kg as-fed basis) ^§^		
Cp (N × 6.25)	160	141
Lysine	11.6	10.3
Methionine	3.3	2.9
Methionine + Cysteine	6.2	5.4
Threonine	6.5	5.6
Tryptophan	2.0	1.7
Leucine	14.5	13.2
Isoleucine	7.0	6.0
Valine	7.5	6.6
Phenylalanine	7.9	6.9

^†^ Diets were formulated based upon the potential of each ISS group for protein deposition according to the NRC Swine model [3]. Provided the following amounts of vitamins and trace minerals (per kg of diet): vitamin A, 10075 IU; vitamin D3,1100 IU; vitamin E, 83 IU; vitamin K (as menadione), 3.7 mg; D-pantothenic acid, 58.5 mg; riboflavin, 18.3 mg; choline, 2209.4 mg; folic acid, 2.2 mg; niacin, 73.1 mg; thiamin, 7.3 mg; pyridoxine, 7.3 mg; vitamin B12, 0.1 mg; D-biotin, 0.4; Cu, 12.6 mg; Fe, 100 mg; Mn, 66.8 mg; Zn, 138.4 mg; Se, 0.3 mg; I, 1.0 mg; S, 0.8 mg; Mg, 0.0622%; Na, 0.0004%; Cl, 0.0336%; Ca, 0.0634%, P, 0.003%; K, 0.0036%. ^§^ Calculated and analyzed crude protein (Cp) and amino acids are standardized ileal digestible (SID) and total basis. * STTD P: Standardized total tract digestible phosphorous.

**Table 2 animals-09-00323-t002:** The impacts of immune system stimulation (ISS) on whole-body nitrogen (N) metabolism in growing pigs ^1^.

Measures	Health Status		SE	*p*-Value
	ISS−	ISS+		
Animals, *n*	5	7		
Final BW, kg BW^0.60^	8.64	8.14	0.083	0.018
N intake, g/kg BW^0.60^/day	2.84	2.00	0.077	0.001
N excretion, g/kg BW ^0.60^/day	0.59	0.58	0.070	0.917
N excretion via ammonia, g/kg BW^0.60^/day	0.05	0.04	0.009	0.814
N excretion via urea, g/kg BW^0.60^/day	0.51	0.56	0.054	0.413
^15^N administered, mg/kg BW^0.60^	7.10	6.79	0.180	0.168
N Flux ^2^, g/kg BW^0.60^/day	12.20	8.65	0.276	0.001
Protein synthesis, g N/kg BW^0.60^/day	11.63	8.10	0.277	0.001
Protein degradation, g N/kg BW^0.60^/day	9.40	6.71	0.319	0.001
Protein retention ^3^, g N/kg BW^0.60^/day	2.28	1.42	0.130	0.001
PD ^4^, g/kg BW^0.60^/day	2.10	1.52	0.151	0.017
N retention:N intake	0.80	0.69	0.040	0.033
Protein synthesis: Protein degradation	1.24	1.19	0.024	0.065
Protein synthesis:Protein retention	5.12	5.87	0.240	0.055

^1^ Data are the least square means ± the largest standard error of mean (SE). Twelve gilts (initial BW 31 ± 4.8 kg) were subjected to one of two health states: immune system stimulated (ISS+; *n* = 7), or healthy control (ISS−; *n* = 5). Repeated i.m. injection of *E. coli* lipopolysaccharide (LPS; 25 and 35 µg/kg BW, given 48 h apart) was used to induce ISS. Pigs in the ISS− group received sterile saline i.m. Diets were formulated to meet daily standardized ileal digestible (SID) amino acid requirements for each ISS group, which were estimated based on the potential of each group for protein deposition according to the NRC Swine model [3]. ^2^ Calculated as the arithmetic mean of urinary urea and ammonia N flux. ^3^ Calculated as the difference between protein synthesis and protein degradation. ^4^ PD, protein deposition, was measured using the nitrogen balance study.

**Table 3 animals-09-00323-t003:** Effects of immune system stimulation (ISS) on nuclei density and muscle fiber composition in different muscles of growing pigs *.

Muscle Fiber Characteristics	Health Status		SE	*p*-Value		
	ISS−	ISS+		ISS	MS	ISS × MS
Nuclei density, mm^2^						
LD	1134	995	76.7	0.018	0.001	0.001
PM	1367 ^a^	1076 ^b^	54.2			
ST	1110	1109	78.0			
SV	1065	1060	34.8			
MHC-I, %						
LD	8.9 ^b^	12.7 ^a^	1.10	0.001	0.001	0.034
PM	21.0	24.6	2.42			
ST	20.8	26.0	4.88			
SV	16.6 ^b^	28.9 ^a^	2.32			
MHC-IIA, %						
LD	17.0	16.5	1.80	0.048	0.001	0.160
PM	27.7	23.1	2.78			
ST	25.7	18.9	2.41			
SV	12.2	13.0	2.00			
MHC-IIX, %						
LD	72.7 ^a^	68.1 ^b^	1.53	0.035	0.01	0.051
PM	50.3	47.1	4.58			
ST	52.2	52.3	6.71			
SV	70.6 ^a^	55.3 ^b^	3.30			

* Data are the least square means ± the largest standard error of mean (SE). Twelve gilts (initial BW 31 ± 4.8 kg) were subjected to one of two health states: immune system stimulated (ISS+; *n* = 7), or healthy control (ISS−; *n* = 5). Repeated i.m. injection of *E. coli* lipopolysaccharide (LPS; 25 and 35 µg/kg BW, given 48 h apart) was used to induce ISS. Pigs in the ISS− group received sterile saline i.m. Diets were formulated to meet daily standardized ileal digestible (SID) amino acid requirements for each ISS group, which were estimated based on the potential of each group for protein deposition according to the NRC Swine model [3]. Pigs were euthanized 72 h post-ISS and muscle samples (MS) were collected immediately following euthanasia. *Longissimus dorsi* (LD), *psoas major* (PM), *semitendinosus* (ST), and *serratus ventralis* (SV). Means marked with superscript letters differ from each other significantly.

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
