# Peer review of "Immune System Stimulation Reduces the Efficiency of Whole-Body Protein Deposition and Alters Muscle Fiber Characteristics in Growing Pigs"

_animals, 2019, doi:10.3390/ani9060323_

Round 1

Reviewer 1 Report

This manuscript simulated the effect of an infection on muscle fibre dynamics in the growing gilt. This study is significant as the grower pig is immunologically naïve and in a phase of rapid skeletal muscle deposition. The quality of the work is impressive, coupling tracer methodologies with histological measures of muscle fibre dynamics. This combination of the tracer methodologies for protein turnover kinetics with the muscle fibre dynamics differentiates this papers from others in the field. The reason for this is that skeletal muscle is comprised of multinucleated cells does not have the same tissue dynamics as proliferative organs such as the gut or liver. Additionally the authors have gone to the effort to characterise muscles comprising different fibre types as they respond differently to stimuli. In addition to the high quality of science displayed the manuscript is of an extremely high standard, being well written and easy to follow. This is a high quality experiment relative to other publications in the field and the authors should be congratulated for their efforts.  Specific comments Table 2 needs to have the column for the error labelled, it appears to have two sets of P-values. For the P-values I recommend using an extra decimal place where results are significant (eg 0.50 for not significant 0.050 instead of 0.05 for significant) 23 "sick" is colloquial, suggest "simulated bacterial infection" or other term. I suggest replacing "sick" with "ill" throughout 25 change hour to hours 29 while it is true that this may affect pork quality, wouldn't compromised protein deposition and growth rates be the primary concern? 219 For nuclei density was anything done to determine if these were myonuclei or was this just a total count. A total count is not very meaningful unless you have an idea what the nuclei were associated with (connective tissue, precursor cells, immune cells etc). Without this context the result may be misleading.

Author Response

Thank you for the comments and suggestions. We found the comments helpful and have modified the paper accordingly, as detailed below. We believe these changes result in an improved manuscript and hope they meet with your approval.

Please see our answers after >>

REVIEWER 1

This manuscript simulated the effect of an infection on muscle fibre dynamics in the growing gilt. This study is significant as the grower pig is immunologically naïve and in a phase of rapid skeletal muscle deposition. The quality of the work is impressive, coupling tracer methodologies with histological measures of muscle fibre dynamics. This combination of the tracer methodologies for protein turnover kinetics with the muscle fibre dynamics differentiates this papers from others in the field. The reason for this is that skeletal muscle is comprised of multinucleated cells does not have the same tissue dynamics as proliferative organs such as the gut or liver. Additionally the authors have gone to the effort to characterise muscles comprising different fibre types as they respond differently to stimuli. In addition to the high quality of science displayed the manuscript is of an extremely high standard, being well written and easy to follow. This is a high quality experiment relative to other publications in the field and the authors should be congratulated for their efforts.

>> Thank you very much for your encouraging words.

 Specific comments

Table 2 needs to have the column for the error labelled, it appears to have two sets of P-values.

>> The corrections have been made. Please see Table 2 in the revised document.

For the P-values I recommend using an extra decimal place where results are significant (eg 0.50 for not significant 0.050 instead of 0.05 for significant)

>> Done. Please see the revised manuscript.

 23 "sick" is colloquial, suggest "simulated bacterial infection" or other term. I suggest replacing "sick" with "ill" throughout

>> Changes have been made throughout the manuscript. Please see line 23 and 28 in the new document.

25 change hour to hours

>> Done. Please see line 25 in the new document.

29 while it is true that this may affect pork quality, wouldn't compromised protein deposition and growth rates be the primary concern?

>> Yes. The results of the current study provide evidence for the negative effects of ISS on lean tissue,both quantitatively and qualitatively. We believe both are equally important.

219 For nuclei density was anything done to determine if these were myonuclei or was this just a total count. A total count is not very meaningful unless you have an idea what the nuclei were associated with (connective tissue, precursor cells, immune cells etc). Without this context the result may be misleading.

>> The results in the current study present the total nuclei count within the muscle fiber. In agreement with the findings of the current study, a number of studies have demonstrated that there is a positive correlation between total nuclei density and cross-sectional area, and thus the muscle mass (Bruusgaard et al.2003, doi: 10.1113/jphysiol.2003.045328; Bruusgaard et al., 201, doi: 10.1073/pnas.0913935107; Hosford et al., 2015, :10.2527/jas2015-9047). The latter provide evidence for reduced protein synthesis in the skeletal muscle, especially large muscles when a reduced total nuclei count is seen. Definitely, further research is needed to determine the effects of ISS on satellite cells and the distribution of nuclei in various compartments of muscle tissue.

Reviewer 2 Report

Dear Authors,

Your submitted paper aims at establishing the influence of Immune System Stimulation (ISS) in 31 kg BW PIC (Pig Improvement Company North America) growing gilts (around 10 weeks of age) on whole-body protein metabolim and muscle fiber characteristics in four different skeletal muscles (longissimus dorsi (LD), serratus ventralis (SV), semitendinosus (SV) and psoas major (PM)). For this, ISS was induced by two intramuscular injections of LPS (Escherichia coli LipoPolySaccharide) 48 h apart and animals were euthanized 72 h after the first injection. Whole- body protein metabolim was determined using the end-product method following the administration of a single oral dose of 15N-glycine prior to the first LPS injection. Control (ISS-, n = 5 animals) and LPS injected (ISS+, n = 7) pigs were all fed an isoenergetic diet according to their nutrient requirements, i.e. ad libitum for ISS- pigs and based on performance variables determined in previous studies for ISS+ pigs, so that ISS- pigs ate less (850 vs 1250 g/d), deposited less protein (60 vs 100 g/d) and were euthanized at a lighter body weight. The nutritional experimental design is physiologically relevant but a third group (ISS- perfed with ISS+) would have been highly informative to separate the effects of reduced Feed Intake (FI) from those of immunopathology per se. The paper is well written and clearly presented with objectives that are in the scope of the targeted journal. However, the study contains some weaknesses that deserve to be addressed to improve for the paper. Major comments along with more minor points are presented below.

Comment 1 : As written in lines 252-253, « Feed intake was used as a co-variate for determining the effect of ISS on muscle fiber characteristics, hematology and measures of blood chemistry Â». I think that it is not relevant because Feed Intake (FI) is imposed by the experimental protocole and its effect is confounded with that of ISS treatment. As discussed lines 486-494, reduced FI is an unavoidable component of the immune response. In the present study, you cannot separate the specific effects of immunopathology from those of reduced FI ; this could have been done by adding a third ISS- group perfed with the ISS+ group, which would have strengthened the experimental design. Consequently, FEED INTAKE should not be used as a co-variable in the statistical analyses and corresponding results and discussion must be revised throughout the paper.

Comment 2 : Monoclonal antibody 10F5 was used to highlight myofibers containing type IIB myosin heavy chain (MHC). Surprisingly, positive type IIB fibers could be identified in the muscles of only two pigs (lines465-466). On the opposite, 48.5% positive type IIB fibers were previously observed in pig longissimus muscle at 3 weeks of age using BF-F3, another monoclonal type IIB MHC antibody (Fazarinc et al., 2017, Animal 11 :164-174). In the present paper, no picture of immunohistochemical stainigs are shown, which should be done at least as supplementary data for the reader to make its own opinion about the quality of immunological stainings. More details should also be given for the number of fields and fibers that were counted to estimate fiber type composition. Fazarinc et al.s paper should be used in the discussion.

Comment 3 : RT-PCR is now a common method to characterize the expression of pig types I, IIA, IIX and IIB MHC at the mRNA level. These data should be added to complete immunohistochemical results and make conclusions on the effects of ISS on fiber type composition more convincing, all the more that the experiment was very short (72 hours), which can be sufficient to induce changes at the mRNA level, whereas a response at the protein level likely needs more time. Metabolic enzyme activities,  such as lactate deshydrogenase, citrate synthase and β-hydroxy-acyl-CoA deshydrogenase could also have been measured to characterize muscle energy metabolism.

Comment 4 : Only 5 ISS- and 7 ISS+ pigs were used, which is extremly low to show significant differences in fiber type composition. Ideally, animals should have been alloted between ISS- and ISS+ groups within litter to minimize individual variability and include the litter effect in the statistical analysis. It is not clear how animals were alloted in the different groups (litter effect ? birth weights ? completely at random ?). In Table 3, when the ISS x MT (Muscle Type) interaction is significant, results should have been analyzed and presented within each muscle to get a clear picture of the differential effects among muscles.

Comment 5 : It is well known that semitendinosus muscle is a highly heterogeneous muscle with a deep red portion (around 40% type I fibers) and a superficial white portion (around 3% type I fibers). In the present study, ST muscle contained 23% type I fibers, which suggests that muscle sampling was carried out in the transition zone between the red and white portions. More details should be given to describe the exact location of muscle sampling within each muscle type (Lines 153-155)

Comment 6 : Because FI of ISS- pigs was reduced  (850 vs 1250 g/d), they exhibited a smaller BW at 72 h following the first LPS injection, which logically explains the reduced muscle fiber cross sectional areas in LD and ST of ISS+ compared to ISS- pigs. Indeed, postnatal muscle growth is related to muscle fiber hypertrophy since the total number of fibers is comonly reported to be definitely fixed before birth.  

Altogether, the effects of ISS on whole body protein metabolism are clearly demontrated, eventhough a third group (ISS- perfed with ISS+) would have been highly informative to separate the effects of reduced FI from those of immunopathology per se. In contrast, results about the effects of ISS on muscle fiber type composition are less convincing and need to be completed.  

MINOR COMMENTS

Lines 82-89 : indicate that gilts were used.

Line 156 : 2 x 2 x 3 cm muscle samples seem to be rather big to get a rapid freezing and good preservation of the muscle structure for good immunohistochemical stainings. Representative pictures of immunohistochemical stainings should be shown, at least as supplementary data.

Line 476 : « general decrease in nuclei density Â» is not appropriate because nuclei density was reduced only in PM.

Line 514 : ERROR. Change « reducing the percentage of slow twitching fibers» to « increasing the percentage of slow twitching fibers Â»

Lines 547, 567, 574, 582, 585, 596 : use abbreviation for journal’s name

Table 2 : ajust column labelings.

END

Author Response

Thank you for the comments and suggestions. We found the comments helpful and have modified the paper accordingly, as detailed below. We believe these changes result in an improved manuscript.

Please see our answers after >>

REVIEWER 2

Dear Authors,

Your submitted paper aims at establishing the influence of Immune System Stimulation (ISS) in 31 kg BW PIC (Pig Improvement Company North America) growing gilts (around 10 weeks of age) on whole-body protein metabolim and muscle fiber characteristics in four different skeletal muscles (longissimus dorsi (LD), serratus ventralis (SV), semitendinosus (SV) and psoas major (PM)). For this, ISS was induced by two intramuscular injections of LPS (Escherichia coli LipoPolySaccharide) 48 h apart and animals were euthanized 72 h after the first injection. Whole- body protein metabolim was determined using the end-product method following the administration of a single oral dose of 15N-glycine prior to the first LPS injection. Control (ISS-, n = 5 animals) and LPS injected (ISS+, n = 7) pigs were all fed an isoenergetic diet according to their nutrient requirements, i.e. ad libitum for ISS- pigs and based on performance variables determined in previous studies for ISS+ pigs, so that ISS- pigs ate less (850 vs 1250 g/d), deposited less protein (60 vs 100 g/d) and were euthanized at a lighter body weight. The nutritional experimental design is physiologically relevant but a third group (ISS- perfed with ISS+) would have been highly informative to separate the effects of reduced Feed Intake (FI) from those of immunopathology per se. The paper is well written and clearly presented with objectives that are in the scope of the targeted journal. However, the study contains some weaknesses that deserve to be addressed to improve for the paper. Major comments along with more minor points are presented below.

Comment 1 : As written in lines 252-253, « Feed intake was used as a co-variate for determining the effect of ISS on muscle fiber characteristics, hematology and measures of blood chemistry Â». I think that it is not relevant because Feed Intake (FI) is imposed by the experimental protocole and its effect is confounded with that of ISS treatment. As discussed lines 486-494, reduced FI is an unavoidable component of the immune response. In the present study, you cannot separate the specific effects of immunopathology from those of reduced FI ; this could have been done by adding a third ISS- group perfed with the ISS+ group, which would have strengthened the experimental design. Consequently, FEED INTAKE should not be used as a co-variable in the statistical analyses and corresponding results and discussion must be revised throughout the paper.

>> The main goal of the current study was to evaluate the effects of ISS on whole-body protein turnover when pigs are fed according to their potential for growth and protein deposition. Pair-feeding a healthy pig is not physiological and results in an inaccurate and unrealistic estimation of protein turnover and efficiency of protein deposition in healthy pigs. Others and we have shown this previously (Rakhshandeh et al., 2014, Brit J Nutr, DOI: 10.1017/S0007114513001955; de Lange 2003, Proceedings of 9th DPP. Edmonton, Alberta, Canada: the University of Alberta. p. 243–261; NRC Swine 2012). Thus, we did not include a pair-fed group in our experimental design to comply with the three R’s principle (Replacement, Reduction, and Refinement) of animal welfare and testing.

In the current study, the muscle fiber characterization was a supplementary measurement to the whole-body protein metabolism study. To our knowledge, this was the first attempt to evaluate the effects of ISS on the muscle fiber characteristics of growing pigs and can be considered a preliminary study on the effects of ISS on these parameters. Obviously further studies are needed to fully elucidate the effects of ISS on muscle fiber characteristics.  Also, the effects of feed intake on muscle fiber characteristics has been studied previously and was not within the scope of the present study. Therefore, the focus of our research was to get an idea of how ISS impacts muscle fiber characteristics.  The reduced feed intake during ISS is inevitable, as we have shown in several studies previously (Rakhshandeh & de Lange, 2012,  doi: 10.1017/S1751731111001522; McGilvray et al., 2019, doi: 10.1093/jas/sky401.).  Thus, it is not irrelevant to use feed intake as a covariate to account for the effects of ISS on immune function and muscle fiber characteristics.  Furthermore, as stated in the M&M we used a reduced model when FI was used as co-variable. Thus, the majority of the data represent the best estimate of the mean without a co-variable. The co-variable effect was only observed on nuclei density.

>> We have corrected data in tables, figures and the text to provide further clarification.  Please see table 3 and lines 247-261 and 314-331 in the revised document.

Comment 2 : Monoclonal antibody 10F5 was used to highlight myofibers containing type IIB myosin heavy chain (MHC). Surprisingly, positive type IIB fibers could be identified in the muscles of only two pigs (lines465-466). On the opposite, 48.5% positive type IIB fibers were previously observed in pig longissimus muscle at 3 weeks of age using BF-F3, another monoclonal type IIB MHC antibody (Fazarinc et al., 2017, Animal 11 :164-174). In the present paper, no picture of immunohistochemical stainigs are shown, which should be done at least as supplementary data for the reader to make its own opinion about the quality of immunological stainings. More details should also be given for the number of fields and fibers that were counted to estimate fiber type composition. Fazarinc et al.s paper should be used in the discussion.

>> Sample images have been provided as a supplementary figure and discussion of the Farzarinc, et al. reference has been added to the text.  Please see lines 468 to 470 in the revised document. The Type IIB result was surprising to us too, as we could see Type IIB MHC in two muscles of the same two pigs, but not in the rest of the pigs. What we understood from these data is that the 10F5 antibody has a positive reaction with Type IIB, when Type IIB is present. Fazarinc et al., did not evaluate the specificity of the 10F5 antibody against other antibodies. Also, we should not forget the differences between pig genetic breeds as far as muscle fiber composition, development, and differentiation.

Comment 3 : RT-PCR is now a common method to characterize the expression of pig types I, IIA, IIX and IIB MHC at the mRNA level. These data should be added to complete immunohistochemical results and make conclusions on the effects of ISS on fiber type composition more convincing, all the more that the experiment was very short (72 hours), which can be sufficient to induce changes at the mRNA level, whereas a response at the protein level likely needs more time. Metabolic enzyme activities,  such as lactate deshydrogenase, citrate synthase and β-hydroxy-acyl-CoA deshydrogenase could also have been measured to characterize muscle energy metabolism.

>> This idea is very important and we agree with the reviewer that RT-PCR would add complementary information to the study.  However, in the current study, we did not measure mRNA levels, as we assumed that changes in protein expression were a reflection of changes in either mRNA transcription or protein translation from mRNA.  Indeed, changes in protein expression are more reflective of changes in the measured parameters of muscle fibers.  Our future studies will focus on characterizing the relationship between mRNA levels and the changes in protein expression that were observed here.

Comment 4 : Only 5 ISS- and 7 ISS+ pigs were used, which is extremly low to show significant differences in fiber type composition. Ideally, animals should have been alloted between ISS- and ISS+ groups within litter to minimize individual variability and include the litter effect in the statistical analysis. It is not clear how animals were alloted in the different groups (litter effect ? birth weights ? completely at random ?). In Table 3, when the ISS x MT (Muscle Type) interaction is significant, results should have been analyzed and presented within each muscle to get a clear picture of the differential effects among muscles.

>> As mentioned above, the main goal of this study was to determine the effects of ISS on the efficiency of protein deposition.  We arrived at the minimum number of biological replications for each ISS group based on a power test.  This information has now been added to the text.  Please see lines 224 to 226.  Obviously, more biological replications will increase statistical confidence.  However, the current number of biological replications used for the muscle fiber portion generated results with relatively high confidence, given that we used a conservative test (Tukey-Kramer) for detecting significant differences.  Thus, the results are of interest.  The Fazarinc et al., (2017, Animal 11: 164-174) study, recommended by the reviewer, also used five biological replications at each time point in their study. 

>> Indeed, the gilts from three litters were allotted to ISS groups, such that at least one representative from each litter was found in each ISS group.  This information has now been added to the text (see lines 86 to 90). 

>>We have added superscripts to data in Table 3 to address the reviewer’s concerns.  Please see the new document.

Comment 5 : It is well known that semitendinosus muscle is a highly heterogeneous muscle with a deep red portion (around 40% type I fibers) and a superficial white portion (around 3% type I fibers). In the present study, ST muscle contained 23% type I fibers, which suggests that muscle sampling was carried out in the transition zone between the red and white portions. More details should be given to describe the exact location of muscle sampling within each muscle type (Lines 153-155)

>> The reviewer’s comment on the composition of type I and II in ST muscle is predominantly applicable to the mature/adult pigs (Lefaucheur et al., 1995, DEVELOPMENTAL DYNAMICS 203:27-41 1995). This characteristic of the ST muscle is less established in growing pigs. Sampling details have been added to the text.  Please see lines 154 to 160.

Comment 6 : Because FI of ISS- pigs was reduced  (850 vs 1250 g/d), they exhibited a smaller BW at 72 h following the first LPS injection, which logically explains the reduced muscle fiber cross sectional areas in LD and ST of ISS+ compared to ISS- pigs. Indeed, postnatal muscle growth is related to muscle fiber hypertrophy since the total number of fibers is comonly reported to be definitely fixed before birth.  

>> This has already been implied in the discussion section.  Please see lines 470 to 492.

Altogether, the effects of ISS on whole body protein metabolism are clearly demontrated, eventhough a third group (ISS- perfed with ISS+) would have been highly informative to separate the effects of reduced FI from those of immunopathology per se. In contrast, results about the effects of ISS on muscle fiber type composition are less convincing and need to be completed.  

 >> As mentioned in our answer to Comment 1, the main goal of the current study was to evaluate the effects of ISS on whole-body protein turnover when pigs are fed according to their potential for growth and protein deposition. Pair-feeding a healthy pig is not physiological and results in an inaccurate and unrealistic estimation of protein turnover and efficiency of protein deposition in healthy pigs. Others and we have shown this previously ((Rakhshandeh et al., 2014, Brit J Nutr, DOI: 10.1017/S0007114513001955; de Lange 2003, Proceedings of 9th DPP. Edmonton, Alberta, Canada: University of Alberta. p. 243–261; NRC Swine 2012; Hewitt et al., 2016, J Anim Sci 98:104). Thus, we did not include a pair-fed group in our experimental design to comply with the three R’s principle (Replacement, Reduction, and Refinement) of animal welfare and testing.  We appreciate the reviewer’s comments about the muscle fiber type composition data and have provided additional information and images to provide a fuller picture of this data.  Clearly, no experimental design is perfect.  However, we believe our data now provides valuable information to the reader, despite there being more to learn about the interaction between muscle fiber characteristics and ISS.

MINOR COMMENTS

Lines 82-89 : indicate that gilts were used.

>> Done.

Line 156 : 2 x 2 x 3 cm muscle samples seem to be rather big to get a rapid freezing and good preservation of the muscle structure for good immunohistochemical stainings. Representative pictures of immunohistochemical stainings should be shown, at least as supplementary data.

>> Done. Please see the supplementary images.

Line 476 : « general decrease in nuclei density Â» is not appropriate because nuclei density was reduced only in PM.

>> We have reanalyzed our data without the co-variable and there is now a strong effect of ISS (the main effect). Thus, we can suggest that nuclei density was, in general, lower in the muscles of ISS pigs.

Line 514 : ERROR. Change « reducing the percentage of slow twitching fibers» to « increasing the percentage of slow twitching fibers Â»

>> Done. Please see line 513 in the new document.

Lines 547, 567, 574, 582, 585, 596 : use abbreviation for journal’s name

>> Done. Please see the reference list in the revised document.

Table 2 : ajust column labelings.

>> Done. Please see table 2 in the revised document.

END
